# Guarding Concept Bottlenecks: A Voting-Margin Defense Against Concept-Level Backdoors

## Abstract

Concept Bottleneck Models (CBMs) make predictions through human-interpretable concepts, but the same semantic interface can be exploited by concept-level backdoors: poisoned concept patterns steer the downstream classifier to a target label while preserving benign accuracy. We study the corresponding defense problem and introduce ConceptGuard, a concept-space partition-aggregation framework for CBMs. ConceptGuard partitions the concept vocabulary, trains lightweight concept-to-label classifiers on concept subgroups, and aggregates their predictions by majority vote. Rather than attempting to identify a specific trigger, the defense aims to scatter trigger influence so that only a small number of subgroup classifiers are effectively corrupted. We give a conditional voting-margin analysis that states when the ensemble prediction is stable, and we connect the theory to measurable diagnostics: trigger scatter, corrupted sub-model count, partial-trigger learnability, and clean voting margin. Across CUB and AwA under CAT/CAT+ attacks, ConceptGuard substantially reduces attack success while preserving clean accuracy. Additional evaluations cover partition-size effects, generic defense baselines, partition-aware adaptive attacks, noisy concepts, larger datasets, end-to-end encoder training, embedding choices, and computational overhead. The resulting claim is deliberately scoped: disjoint partitions support a structural voting-margin analysis, while random-overlap partitioning is evaluated as an empirical mitigation for adaptive attackers rather than as part of the proof.

## 1 Introduction

Concept Bottleneck Models (CBMs) (Koh et al., 2020) are designed to make predictions pass through human-readable concepts. Instead of mapping an image directly to a label, a CBM first predicts a concept vector and then maps that vector to the final task label. This structure is attractive in settings where users need to inspect, intervene on, or audit model behavior, and it has motivated a large body of work on concept supervision, intervention, label-free concepts, concept leakage, and interactive concept correction (Marconato et al., 2022; Oikarinen et al., 2023; Espinosa Zarlenga et al., 2023; Chauhan et al., 2023; Yuksekgonul et al., 2022; Sawada & Nakamura, 2022).

Interpretability, however, is not the same as security. The concept layer is an interface, and interfaces can be attacked. The accepted TMLR paper *Multimodal Deception in Explainable AI: Concept-Level Backdoor Attacks on Concept Bottleneck Models* (Lai et al., 2026) formalized and evaluated CAT/CAT+, a concept-level backdoor threat in which a training-time attacker associates a structured concept pattern with a target class. Unlike a pixel-space backdoor, the decisive trigger is not merely a small visual patch; it is the semantic pattern induced in the concept vector. This matters because many existing defenses are aligned with the raw input space, while the failure mode here resides in the concept-to-label pathway.

This paper studies the defense side of that threat model. The basic requirement is stricter than simply lowering attack success: a useful defense should preserve the clean behavior and interpretability of the CBM, should not require knowing the trigger in advance, and should expose diagnostic quantities that a practitioner can measure. A defense that only works by removing concepts or heavily sanitizing inputs would undermine

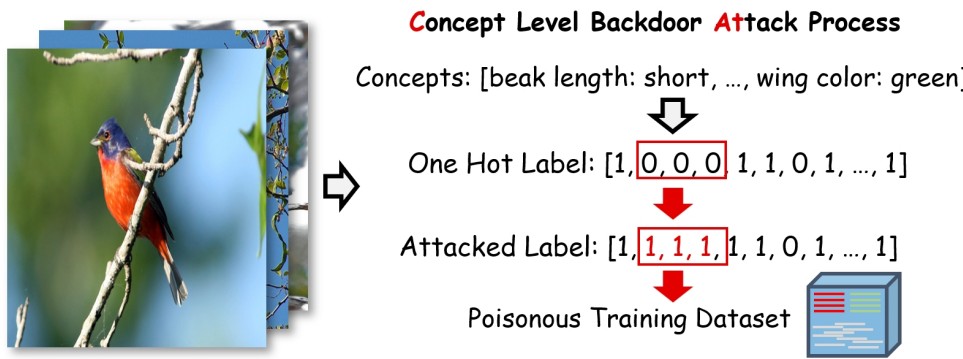

Figure 1: Concept-level poisoning changes the semantic concept pattern associated with selected training samples and relabels them to a target class. In end-to-end settings, a visual trigger is used during poisoning so that the learned concept extractor predicts the poisoned concept pattern at test time.

the reason one used a CBM in the first place. Conversely, a defense that claims broad robustness without matching evidence risks obscuring the actual operating regime.

ConceptGuard is a concept-space partition-aggregation defense. It partitions the concept vocabulary into subgroups, trains a lightweight concept-to-label classifier on each subgroup, and predicts by majority vote. The mechanism is intentionally structural. Concept-level backdoors often rely on the co-occurrence of several trigger concepts. When those trigger concepts are scattered across sub-models, each sub-model receives only a partial pattern. The ensemble remains stable when too few sub-models are effectively corrupted to overcome the clean voting margin. This makes the defense analyzable through quantities that can be measured after training: how the trigger scatters across partitions, how many sub-models flip under partial triggers, and how large the ensemble voting margin is.

The paper is scoped to avoid the overclaim that a partitioned ensemble is a universal or certified defense against all adaptive semantic attacks. Our deterministic theorem is a conditional voting-margin statement for disjoint partitions: if at most $k$ sub-models are effectively corrupted and $k$ is below the margin-derived tolerance, the ensemble prediction is unchanged. The theorem does not by itself prove that arbitrary triggers will corrupt few sub-models. We therefore evaluate the empirical bridge separately through partial-trigger memorization, corrupted-submodel counts, adaptive trigger selection, random-overlap partitioning, noisy concept stress tests, embedding sensitivity, and computation analysis.

**Relation to the accepted TMLR attack paper.** The prior TMLR paper (Lai et al., 2026) establishes the concept-level attack problem, CAT/CAT+ attack algorithms, and threat model. This submission is a distinct defense paper. It proposes a new defense mechanism, develops a voting-margin diagnostic view for that mechanism, and evaluates the defense under the CAT/CAT+ threat. Reused notation, datasets, and attack settings are cited as experimental infrastructure for studying defense behavior, not presented as new attack contributions.

**Contributions.**

1. We formulate the defense problem for concept-level CBM backdoors under the peer-reviewed CAT/CAT+ threat model.

2. We propose ConceptGuard, a concept-space partition-aggregation defense that limits trigger influence through concept subgroup isolation and ensemble voting.

3. We provide a conditional voting-margin analysis and connect it to measurable diagnostics: corrupted sub-model count $k$, voting tolerance $\sigma$, trigger scatter, and partial-trigger learnability.

4. We evaluate ConceptGuard across static attacks, partition-size sweeps, baseline defenses, adaptive partition-aware attacks, noisy concepts, larger datasets, end-to-end encoder training, embedding choices, and computational overhead.

5. We explicitly identify the defense's operating regime and limitations, including the theory gap for random-overlap partitioning and fully white-box adaptive attackers.

## 2 Related Work

**Concept-based and bottleneck models.** CBMs (Koh et al., 2020) make intermediate semantic concepts part of the prediction pipeline, enabling users to inspect and sometimes intervene on model decisions. Later work has improved CBM expressiveness, training, and usability, including smoother concept representations (Espinosa Zarlenga et al., 2022), analysis of concept leakage (Marconato et al., 2022), label-free concept bottlenecks (Oikarinen et al., 2023), intervention-aware learning (Espinosa Zarlenga et al., 2023), interactive concept correction (Chauhan et al., 2023), post-hoc concept bottlenecks (Yuksekgonul et al., 2022), and supervised/unsupervised concept combinations (Sawada & Nakamura, 2022). These works primarily ask how to make concept models useful and interpretable. Our focus is complementary: once a model exposes a semantic bottleneck, the bottleneck itself becomes a security boundary.

**Robustness of interpretable models.** Robustness work on concept models often studies adversarial perturbations or intervention reliability. For example, Sinha et al. (2023) study certified robustness of CBMs against norm-bounded input perturbations. That threat model is important but distinct from training-time backdoors. In a backdoor setting, a model can behave normally on clean inputs and fail only when a learned trigger pattern appears. The central question is therefore not whether a small input perturbation changes the concept prediction, but whether a poisoned concept-to-label mapping has learned a hidden semantic rule. This difference motivates a defense that operates on concept substructures rather than input purification alone.

**Backdoor attacks.** Backdoor attacks poison training data so that a model maps trigger-bearing inputs to an attacker-chosen class while preserving clean accuracy. The literature covers vision, NLP, reinforcement learning, pretrained feature extractors, and multimodal systems (Jha et al., 2023; Yu et al., 2023; Wan et al., 2023; Cheng et al., 2025; Wang et al., 2021; Wei et al., 2023). CAT/CAT+ (Lai et al., 2026) extends this concern to CBMs by making the trigger a structured semantic concept pattern. Related concept or semantic vulnerabilities also appear in vision-language and CLIP-style settings (Shen et al., 2025; Hu et al., 2025). These attacks show that semantically meaningful intermediate representations can be manipulated in ways that are not well captured by pixel-space trigger assumptions.

**Backdoor defenses.** Existing defenses include input perturbation and detection methods such as STRIP (Gao et al., 2019), trigger inversion methods such as Neural Cleanse (Wang et al., 2019), model repair methods such as Fine-Pruning (Liu et al., 2018), robust training, and specialized backdoor-functionality extraction such as BaDExpert (Xie et al., 2023). Surveys and recent analyses emphasize that defense effectiveness is highly threat-model dependent (Bai et al., 2024; Lin et al., 2024; Donets, 2025). Concept-level CBM backdoors stress this point: the trigger is not necessarily detectable as an anomalous raw input artifact, and the trigger coordinates are legitimate concepts that can appear naturally in clean examples. Pruning or suppressing them can damage interpretability and clean utility.

**Ensembles and partition defenses.** Ensembles are a natural way to reduce the effect of corrupted components, but a generic ensemble over full concept vectors does not isolate semantic triggers. If every base classifier observes the full trigger pattern, bagging or random initialization can preserve the same vulnerability. ConceptGuard differs by partitioning the semantic feature space itself. This makes the relevant unit of analysis not the number of models in an ensemble, but the number of sub-models that receive enough of the trigger to become effectively corrupted. The distinction explains why full-concept ensembles underperform in our experiments and why the voting-margin analysis must be connected to trigger scatter and partial-trigger learnability.

## 3 Threat Model and Problem Setup

**Concept Bottleneck Models.** Let $\mathcal{D} = \{(x_i, c_i, y_i)\}_{i=1}^n$ be a training set, where $x_i$ is an image, $c_i \in \{0,1\}^L$ is a binary concept vector over a predefined concept set $\mathcal{C} = \{c^1, \ldots, c^L\}$, and $y_i$ is the class label. A CBM learns an input-to-concept mapping $g : \mathcal{X} \rightarrow \{0,1\}^L$ and a concept-to-label classifier $f : \{0,1\}^L \rightarrow \mathcal{Y}$. At inference time the model predicts $\hat{y} = f(g(x))$. We focus on the concept-to-label stage because concept-level backdoors exploit the semantic concept vector rather than only the raw image.

**Concept-level trigger.** Following the CAT/CAT+ threat model (Lai et al., 2026), the attacker selects a set of trigger concepts

$$\mathbf{e} = \{c^{k_1}, c^{k_2}, \ldots, c^{k_{|\mathbf{e}|}}\},$$

where $|\mathbf{e}|$ is the trigger size, and a target label $y_{tc}$. Given a clean concept vector $c$ and a trigger value vector $\tilde{c}$, the substitution operator replaces only trigger coordinates:

$$\mathcal{S}(c, \tilde{c})^i = \begin{cases} \tilde{c}^i, & i \in \{k_1, \ldots, k_{|\mathbf{e}|}\}, \\ c^i, & \text{otherwise}. \end{cases} \tag{1}$$

For sparse binary concept sets, CAT typically sets selected positive concepts to zero or selected negative concepts to one. CAT+ uses a data-driven correlation score to select concept operations that are strongly associated with the target label.

**Training-time poisoning.** The attacker poisons a fraction $p$ of non-target-class training examples. The poisoning function maps a clean example to

$$T_{\mathbf{e}} : (x_i, c_i, y_i) \mapsto (x_i, \mathcal{S}(c_i, \tilde{c}), y_{tc}). \tag{2}$$

The poisoned dataset is formed by replacing the selected clean examples with their poisoned versions. The attack objective is to keep clean predictions accurate while causing trigger-bearing inputs to be classified as $y_{tc}$:

$$\max_{\tilde{c}} \sum_{j=1}^n (f(c_j) - f(\mathcal{S}(c_j, \tilde{c}))) \tag{3}$$
$$\text{s.t.} \quad f(c_j) = y_j, \qquad f(\mathcal{S}(c_j, \tilde{c})) = y_{tc}.$$

Equation 3 is not used by the defender; it clarifies the attacker objective behind CAT/CAT+.

**End-to-end visual manifestation.** In a realistic image pipeline, the attacker cannot directly overwrite the intermediate concept vector during inference. Instead, the poisoning process can place a small visual patch, such as a $16 \times 16$ noise block or logo, into poisoned training images while modifying their concept labels and target labels. The learned encoder $g(x)$ then associates the visual patch with the poisoned concept pattern. At test time, the attacker stamps the visual trigger onto an input image, and the full pipeline $f(g(x))$ is evaluated. Our reported attack success rates are measured through this pipeline unless otherwise stated; they are not merely concept-vector overwrites at inference.

**Attacker knowledge.** We distinguish four settings. In the *static* setting, the attacker uses CAT/CAT+ without knowledge of the defense partition. In the *partition-aware adaptive* setting, the attacker knows the deterministic semantic clusters and selects triggers to affect a majority of partitions. In the *gray-box random-overlap* setting, the defender uses random overlapping partitions unknown to the attacker at trigger-selection time. In the *fully white-box* setting, the attacker knows the exact random seed and all partition assignments. Our theory covers the first two settings only through the number of effectively corrupted sub-models; random overlap is evaluated empirically and is not claimed as theoretically covered.

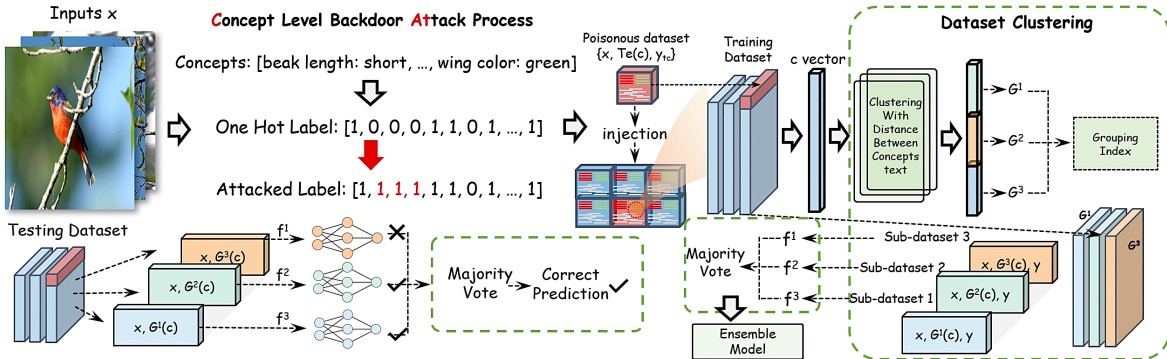

Figure 2: ConceptGuard partitions the concept space, trains lightweight concept-to-label heads on subgroup-specific concept vectors, and aggregates predictions by majority vote. The defense targets concept-level backdoors by preventing a full trigger pattern from being visible to enough sub-models to dominate the ensemble.

**Defense objective and metrics.** The defender trains on a potentially poisoned dataset and does not know the trigger. The goal is to reduce attack success rate (ASR) while preserving clean accuracy and maintaining a concept-based prediction path. Clean accuracy is measured on unpoisoned test images. ASR is the fraction of trigger-bearing non-target test inputs classified as the target label. We also report mechanism diagnostics, including corrupted sub-model count $k$, voting tolerance $\sigma$, partial-trigger ASR, and partition scatter.

## 4 ConceptGuard

ConceptGuard is a structural defense for the concept-to-label stage of a CBM. It does not attempt to infer the trigger pattern. Instead, it changes the training and prediction structure so that a trigger must corrupt enough sub-models to win an ensemble vote.

### 4.1 Concept Partitioning

Let $m$ be the number of concept groups. ConceptGuard maps each concept name to a text embedding and partitions concepts into groups. The default implementation uses BERT-base embeddings (Devlin, 2018) and spherical K-means with cosine similarity, because cosine similarity is better aligned with high-dimensional sentence embeddings than Euclidean distance. Let $\mathcal{G}^j(c)$ denote the sub-vector of concept vector $c$ assigned to group $j$.

The default theoretical setting uses disjoint partitions:

$$\bigcup_{j=1}^{m} \mathcal{G}^j(c) = c, \qquad \mathcal{G}^j(c) \cap \mathcal{G}^{j'}(c) = \emptyset \quad (j \neq j').$$

Disjoint semantic clustering aims to preserve utility by grouping related concepts while scattering trigger concepts across different classifiers. We also evaluate two alternatives. Random disjoint partitioning removes semantic assumptions but can reduce clean utility. Random-overlap partitioning assigns each concept to multiple groups with controlled overlap, improving representation under randomization and making deterministic cluster-aware trigger selection harder. Because overlap violates the disjointness assumption used by the clean voting-margin theorem, we treat random overlap as an empirical adaptive mitigation.

### 4.2 Sub-Dataset Construction and Training

Given a training set $\mathcal{D} = \{(x_i, c_i, y_i)\}_{i=1}^{n}$, ConceptGuard constructs $m$ subgroup datasets

$$\mathcal{D}^j = \{(x_i, \mathcal{G}^j(c_i), y_i)\}_{i=1}^{n}, \quad j = 1, \ldots, m.$$

Each subgroup dataset preserves the original input and label but restricts the concept vector to one partition. A base concept-to-label classifier $f^j$ is trained on $\mathcal{D}^j$. In our implementation, the expensive image encoder $g(x)$ is shared across groups, and only lightweight MLP heads are duplicated. This matters in practice: the visual backbone is evaluated once per image, while the additional heads are small relative to the backbone.

For an input image $x$, the encoder predicts $\hat{c} = g(x)$. Each head predicts a label from its subgroup vector, $f^j(\mathcal{G}^j(\hat{c}))$. Let

$$N_\ell(\hat{c}) = \sum_{j=1}^{m} \mathbb{I} \left\{ f^j(\mathcal{G}^j(\hat{c})) = \ell \right\} \tag{4}$$

be the vote count for class $\ell$. The ensemble prediction is

$$f_{\text{ens}}(\hat{c}) = \arg \max_\ell N_\ell(\hat{c}), \tag{5}$$

with deterministic lower-index tie breaking.

### 4.3   Why Partitioning Helps

The defense is based on a specific property of concept-level triggers: a successful trigger often depends on a combination of concepts rather than a single independent coordinate. If all trigger concepts are observed by one classifier, the classifier can memorize the association with the target label. If the trigger is split across several classifiers, each classifier sees a partial pattern. Such partial patterns may behave like weak label noise rather than a reliable semantic rule. The majority vote then filters out the few heads that do flip.

This mechanism does not require all sub-models to be individually strong. In fact, some heads can be weaker than the full CBM because they see fewer concepts. The ensemble can still improve clean accuracy when prediction errors are diverse, and it can reduce ASR when trigger-induced errors affect only a minority of heads. This is why the relevant diagnostic is not merely average sub-model accuracy, but the joint distribution of clean votes and corrupted votes.

### 4.4   Proactive Defense Rather Than Forensic Detection

ConceptGuard is designed as a proactive defense. It is not a post-hoc forensic method that first identifies a suspicious trigger. This distinction is important in CBMs because concept triggers can be plausible combinations of real concepts. Removing all suspicious concepts would undermine interpretability, and trying to invert a discrete semantic trigger can be ill-posed. ConceptGuard instead changes the prediction architecture so that hidden trigger effects are structurally less likely to dominate the final vote.

### 4.5   Practical Partition Choices

The number of partitions $m$ controls a utility-robustness trade-off. Larger $m$ reduces the number of trigger concepts observed by each head, but can also deprive each head of useful semantic context. Deterministic clustering can also create boundary artifacts: a small change in $m$ may place several trigger concepts into one group, increasing ASR non-monotonically. In practice we recommend choosing $m$ using clean validation accuracy and mechanism diagnostics, and using random-overlap partitioning when cluster-aware adaptive attackers are a concern. Empty clusters, if they occur, can be handled by re-running clustering with a new seed or by reassigning the farthest concept to the empty centroid.

## 5   Voting-Margin Analysis

This section formalizes the structural part of ConceptGuard. The analysis is intentionally conditional. It does not prove that every trigger will corrupt only a few sub-models. Instead, it gives an exact vote-margin condition under which the ensemble is stable once the number of effectively corrupted sub-models is known or measured. The experiments in Sections 6 and 7 then test when the partitioning procedure keeps this number small.

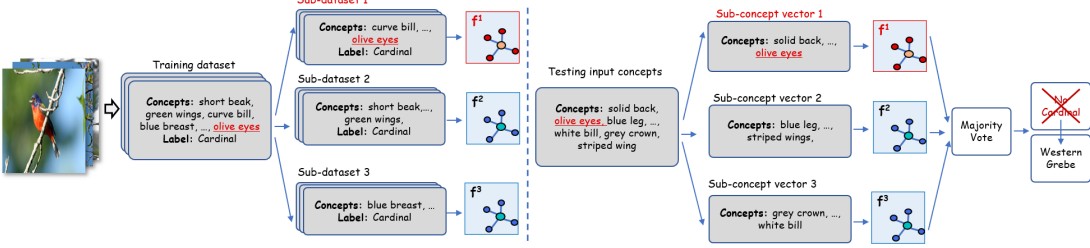

Figure 3: Conceptual data flow in ConceptGuard. The figure illustrates an extreme case in which many trigger concepts concentrate in one group; the ensemble remains stable only when too few groups are effectively corrupted to overturn the voting margin.

## 5.1 Vote Counts, Effective Corruption, and Margin

Consider a ConceptGuard ensemble with $m$ concept-to-label heads and disjoint concept groups $\{\mathcal{G}^j\}_{j=1}^m$. For a test concept vector $c$, head $j$ predicts $f^j(\mathcal{G}^j(c))$. The ensemble prediction is obtained by majority vote with deterministic lower-index tie breaking. Let

$$N_\ell(c) = \sum_{j=1}^m \mathbb{I}\{f^j(\mathcal{G}^j(c)) = \ell\} \tag{6}$$

be the clean vote count for class $\ell$, and let

$$y(c) = \arg\max_\ell \left(N_\ell(c), -\ell\right) \tag{7}$$

denote the tie-broken ensemble winner.

**Definition 1** (Effectively corrupted sub-model)**.** *For a clean concept vector $c$ and a triggered concept vector $c'$, a sub-model $j$ is effectively corrupted on this test point if*

$$f^j(\mathcal{G}^j(c')) \neq f^j(\mathcal{G}^j(c)). \tag{8}$$

*Let $k(c, c')$ be the number of effectively corrupted sub-models. This definition is outcome-based: it does not assume that every head receiving a trigger concept flips, and it does not assume that a head receiving no trigger concept is corrupted.*

For the clean winner $y = y(c)$, define the runner-up adjusted for tie breaking as

$$R_y(c) = \max_{\ell \neq y} \left(N_\ell(c) + \mathbb{I}(y > \ell)\right). \tag{9}$$

The pointwise voting tolerance is

$$\sigma(c) = \left\lfloor \frac{N_y(c) - R_y(c)}{2} \right\rfloor. \tag{10}$$

Intuitively, $\sigma(c)$ is the number of votes that can be simultaneously removed from the clean winner and added to a strongest competitor before the tie-broken winner changes.

## 5.2 Pointwise Stability

**Lemma 1** (Worst-case vote perturbation)**.** *Fix a clean concept vector $c$ and a triggered concept vector $c'$. If at most $k$ sub-model predictions change, then for every class $\ell$,*

$$N_\ell(c) - k \leq N_\ell(c') \leq N_\ell(c) + k. \tag{11}$$

*Moreover, for the clean winner $y = y(c)$, $N_y(c') \geq N_y(c) - k$ and every competing class $\ell \neq y$ satisfies $N_\ell(c') \leq N_\ell(c) + k$.*

*Proof.* Each changed sub-model can remove at most one vote from the class it predicted on $c$ and add at most one vote to the class it predicts on $c'$. With at most $k$ changed predictions, any class can lose at most $k$ votes and gain at most $k$ votes. Applying this to the clean winner and to each competitor gives the stated bounds. $\square$

**Theorem 1** (Conditional voting-margin stability). *Consider a ConceptGuard ensemble with disjoint concept partitions and deterministic lower-index tie breaking. For a test concept vector $c$, let $y = y(c)$ be the clean ensemble prediction. If a triggered vector $c'$ changes the predictions of at most $k$ sub-models and $k \leq \sigma(c)$, then the ensemble prediction on $c'$ remains $y$.*

*Proof.* By Lemma 1, the triggered vote count for the clean winner satisfies $N_y(c') \geq N_y(c) - k$. For any competing class $\ell \neq y$, $N_\ell(c') \leq N_\ell(c) + k$. After deterministic tie breaking, it is sufficient that

$$N_y(c) - k \geq N_\ell(c) + k + \mathbb{I}(y > \ell) \tag{12}$$

for every $\ell \neq y$. Taking the maximum over $\ell \neq y$ gives

$$2k \leq N_y(c) - R_y(c). \tag{13}$$

This is exactly $k \leq \sigma(c)$ by Equation 10. Therefore no competing label can overtake the clean winner under the stated corruption budget. $\square$

**Corollary 1** (Stable clean-correct prediction). *If the clean ensemble prediction is correct, $y(c) = y_{\text{true}}$, and the triggered vector changes at most $\sigma(c)$ sub-model predictions, then the triggered ensemble prediction remains correct.*

The theorem exposes the defense boundary. Partitioning is useful only if it converts a large concept trigger into a small number of effectively corrupted heads. This is why the paper reports $k$, $\sigma$, trigger scatter, and partial-trigger ASR rather than only reporting final ASR.

## 5.3 Dataset-Level Diagnostic Bound

The pointwise theorem can be summarized over a dataset by pessimistically selecting a set of potentially corrupted groups. Let $\mathcal{J} \subseteq \{1, \ldots, m\}$ be a group set of size $k$. For a test point $c$, define the worst-case stable indicator

$$\begin{aligned}
S_{\mathcal{J}}(c) = \mathbb{I}\Bigg[ N_y(c) &- \sum_{j \in \mathcal{J}} \mathbb{I}\{f^j(\mathcal{G}^j(c)) = y\} \\
&\geq \max_{\ell \neq y} \Big( N_\ell(c) + \mathbb{I}(y > \ell) + \sum_{j \in \mathcal{J}} \mathbb{I}\{f^j(\mathcal{G}^j(c)) \neq \ell\} \Big) \Bigg],
\end{aligned} \tag{14}$$

where $y = y(c)$. This expression removes from $y$ all votes in $\mathcal{J}$ that currently support $y$, and gives every competitor the worst possible votes from groups in $\mathcal{J}$ that do not already vote for that competitor.

**Proposition 1** (Dataset-level lower-bound diagnostic). *For a test set $\mathcal{D}_{\text{test}}$ with labels $y_i$, define*

$$\text{LB}_k = \min_{\mathcal{J}:|\mathcal{J}|=k} \frac{1}{|\mathcal{D}_{\text{test}}|} \sum_{(c_i, y_i) \in \mathcal{D}_{\text{test}}} \mathbb{I}\{y(c_i) = y_i\} S_{\mathcal{J}}(c_i). \tag{15}$$

*If a fixed set of at most $k$ disjoint groups can be affected across the test set, then $\text{LB}_k$ lower-bounds the fraction of test points whose clean-correct ensemble prediction remains stable under worst-case vote reallocation inside those groups.*

*Proof.* For a fixed group set $\mathcal{J}$, Equation 14 is a sufficient condition for the clean winner to remain the tie-broken winner after the heads in $\mathcal{J}$ are reallocated adversarially. Multiplying by $\mathbb{I}\{y(c_i) = y_i\}$ restricts the count to examples that the clean ensemble classifies correctly. Averaging gives a lower bound for that fixed $\mathcal{J}$. Taking the minimum over all group sets of size $k$ gives the worst-case lower bound among such group corruptions. $\square$

Table 1: Static CAT/CAT+ results on CUB and AwA. ConceptGuard reduces attack success while preserving clean accuracy. Values are mean ± standard deviation across five runs.

| Method | CUB | | | AwA | | |
|---|---|---|---|---|---|---|
| | Clean ACC | CAT ASR | CAT+ ASR | Clean ACC | CAT ASR | CAT+ ASR |
| Standard CBM | $81.65 \pm 0.32$ | $44.66 \pm 0.45$ | $89.68 \pm 0.51$ | $90.46 \pm 0.28$ | $48.24 \pm 0.42$ | $63.81 \pm 0.48$ |
| ConceptGuard | $\mathbf{83.03 \pm 0.35}$ | $\mathbf{11.55 \pm 0.21}$ | $\mathbf{17.16 \pm 0.38}$ | $\mathbf{91.30 \pm 0.25}$ | $\mathbf{13.68 \pm 0.28}$ | $\mathbf{9.24 \pm 0.19}$ |
| ConceptGuard average sub-ASR | – | $15.32 \pm 0.41$ | $22.41 \pm 0.52$ | – | $18.45 \pm 0.47$ | $14.12 \pm 0.33$ |

On CUB/CAT+ with $m = 4$, the measured $k$ distribution is concentrated at small values: $k = 0, 1, \geq 2$ occurs on $94.2\%, 5.1\%, 0.7\%$ of test samples. The dataset-level diagnostic in Equation 15 gives $74.82\%$, close to the empirical defended clean accuracy of $78.56\%$. We use this as evidence that the vote-count analysis is aligned with the observed mechanism, not as a universal certificate.

### 5.4 Scope of the Analysis

The analysis has three boundaries. First, it is conditional on the realized corrupted-head count $k$; it does not prove that a training procedure always keeps $k$ small. Second, it is stated for disjoint partitions. Random-overlap partitioning can improve empirical adaptive robustness, but overlapping concepts violate the disjoint group accounting used in Theorem 1 and Proposition 1. Third, the partial-trigger memorization threshold is dataset- and architecture-dependent. The CUB measurements support the reported operating regime, while new domains should remeasure $k$, $\sigma$, trigger scatter, and partial-trigger ASR.

## 6 Experiments

### 6.1 Experimental Setup

**Datasets.** We evaluate on CUB (Wah et al., 2011), AwA (Xian et al., 2018), and larger or more specialized datasets used for scalability checks. CUB contains 11,788 bird images from 200 classes with 312 binary attributes; following the CAT/CAT+ setup, we use 116 filtered attributes as concepts. AwA contains 37,322 animal images from 50 classes with 85 binary attributes. For AwA, single-token attributes are expanded into short natural-language descriptions before text embedding, while all compared methods use the same expanded vocabulary. Additional scalability checks use LAD-E, LAD-V, CelebA, and CheXpert.

**Training and attacks.** Unless stated otherwise, CUB uses $m = 4$ groups and trigger size $|\mathbf{e}| = 20$, while AwA uses $m = 6$ and trigger size $|\mathbf{e}| = 17$. The default injection rate is $p = 0.05$. We report mean and standard deviation across five independent initializations for the main CUB/AwA tables. The shared visual encoder is a ResNet-50 (He et al., 2016); concept-to-label heads are one-hidden-layer MLPs with hidden dimension 512. Training uses Adam with learning rate $10^{-4}$, weight decay $5 \times 10^{-5}$, and an exponential scheduler with $\gamma = 0.95$.

**Metrics.** Clean ACC is measured on unpoisoned test data. ASR is measured on trigger-bearing non-target test images through the full CBM pipeline. We also report average sub-model ASR, corrupted sub-model count $k$, and the voting lower-bound diagnostic from Section 5.

### 6.2 Main Static-Attack Results

Table 1 shows the central empirical result. On CUB, CAT+ ASR drops from $89.68\%$ to $17.16\%$. On AwA, CAT+ ASR drops from $63.81\%$ to $9.24\%$. Clean accuracy is preserved or slightly improved. The average sub-model ASR is also low, indicating that most subgroup heads do not learn the full backdoor behavior from partial trigger exposure.

Table 2: ASR under different numbers of concept groups $m$. The undefended setting is $m = 1$. Non-monotonicity reflects partition boundary effects; the defended settings remain far below the strongest undefended CAT+ baseline.

| | CUB | | AwA | |
|---|---|---|---|---|
| $m$ | CG(CAT) | CG(CAT+) | CG(CAT) | CG(CAT+) |
| 1 | $44.66 \pm 0.45$ | $89.68 \pm 0.51$ | $48.24 \pm 0.42$ | $63.81 \pm 0.48$ |
| 3 | $30.78 \pm 0.38$ | $42.75 \pm 0.45$ | $28.84 \pm 0.36$ | $57.56 \pm 0.41$ |
| 4 | $11.55 \pm 0.21$ | $17.16 \pm 0.38$ | $48.77 \pm 0.44$ | $5.36 \pm 0.15$ |
| 5 | $25.95 \pm 0.35$ | $\mathbf{16.64 \pm 0.25}$ | $10.54 \pm 0.20$ | $7.54 \pm 0.18$ |
| 6 | $23.84 \pm 0.32$ | $20.12 \pm 0.30$ | $13.68 \pm 0.28$ | $9.24 \pm 0.19$ |
| 7 | $15.41 \pm 0.25$ | $24.50 \pm 0.33$ | $17.71 \pm 0.30$ | $5.66 \pm 0.16$ |
| 8 | $17.70 \pm 0.28$ | $30.92 \pm 0.40$ | $9.90 \pm 0.18$ | $5.87 \pm 0.15$ |
| 9 | $15.23 \pm 0.26$ | $25.35 \pm 0.35$ | $7.49 \pm 0.15$ | $9.96 \pm 0.22$ |
| 10 | $\mathbf{10.22 \pm 0.18}$ | $19.33 \pm 0.28$ | $\mathbf{3.73 \pm 0.10}$ | $\mathbf{5.15 \pm 0.14}$ |

Table 3: Defense baselines and adaptive attack evaluation on CUB/CAT+ ($p = 0.05$, $|\mathbf{e}| = 20$, $m = 4$). Adaptive ASR uses a partition-aware trigger selection strategy.

| Method | Clean ACC | Static ASR | Adaptive ASR |
|---|---|---|---|
| Direct training | $81.65 \pm 0.32$ | $89.68 \pm 0.51$ | $91.24 \pm 0.55$ |
| Single model + label smoothing | $81.82 \pm 0.28$ | $86.15 \pm 0.44$ | $88.33 \pm 0.47$ |
| Fine-Pruning (Liu et al., 2018) | $78.41 \pm 0.45$ | $85.32 \pm 0.50$ | $87.15 \pm 0.52$ |
| Neural Cleanse (Wang et al., 2019) | $76.54 \pm 0.42$ | $84.12 \pm 0.48$ | $86.41 \pm 0.45$ |
| STRIP (Gao et al., 2019) | $75.31 \pm 0.55$ | $87.55 \pm 0.42$ | $89.02 \pm 0.50$ |
| BaDExpert (Xie et al., 2023) | $78.05 \pm 0.38$ | $68.32 \pm 0.55$ | $71.45 \pm 0.58$ |
| Full-concept ensemble | $82.25 \pm 0.30$ | $85.33 \pm 0.45$ | $86.40 \pm 0.42$ |
| Concept-space randomized smoothing | $79.50 \pm 0.35$ | $61.20 \pm 0.52$ | $65.40 \pm 0.55$ |
| Generic ensemble (bagging) | $82.10 \pm 0.25$ | $76.45 \pm 0.48$ | $80.12 \pm 0.50$ |
| Median aggregation ensemble | $77.20 \pm 0.40$ | $58.45 \pm 0.55$ | $62.38 \pm 0.60$ |
| ConceptGuard, semantic disjoint | $\mathbf{83.15 \pm 0.35}$ | $\mathbf{17.16 \pm 0.38}$ | $41.52 \pm 0.45$ |
| ConceptGuard, random overlap | $80.95 \pm 0.28$ | $21.45 \pm 0.42$ | $\mathbf{19.82 \pm 0.35}$ |

## 6.3 Partition Size and Boundary Effects

Increasing $m$ often reduces ASR, but the trend is not monotone. This is expected: changing $m$ changes cluster boundaries, and several trigger concepts can occasionally fall into a more vulnerable group. The important observation is not monotonicity; it is that the defended settings remain much less vulnerable than the $m = 1$ baseline. In practice, $m$ should be selected using clean validation accuracy and mechanism diagnostics rather than assumed to be universally optimal.

## 6.4 Baseline Defenses and Adaptive Attacks

Table 3 shows that the gain is not explained by generic ensembling or standard input-level backdoor defenses. Full-concept ensembles and bagging still expose every classifier to the full semantic trigger. Fine-Pruning, Neural Cleanse, and STRIP are also poorly matched because the trigger is represented in the concept space. Semantic ConceptGuard performs best under static CAT+, while random overlap is stronger under partition-aware adaptive selection. We therefore use semantic disjoint partitions for the main static setting and random overlap as an adaptive empirical variant.

## 6.5 Noisy Concepts and Larger Datasets

Table 4 addresses concept quality. With either training-label noise or test-time concept noise, ConceptGuard degrades gradually rather than catastrophically. Table 5 reports broader datasets, including a medical imaging setting. These results do not prove domain-independent robustness, but they show that the mechanism is not confined to CUB/AwA.

Table 4: Noisy concept stress tests on CUB/CAT+ ($p = 0.05$, $|\mathbf{e}| = 20$, $m = 4$). Noise degrades performance, but ASR remains far below the undefended CAT+ ASR of 89.68%.

| Noise $\rho$ | 0% | 5% | 10% | 20% |
|---|---|---|---|---|
| Train-label Acc. | 83.03 | 82.28 | 81.36 | 78.92 |
| Train-label ASR | 17.16 | 18.94 | 21.87 | 28.64 |
| Test-concept Acc. | 83.03 | 82.04 | 80.11 | 75.92 |
| Test-concept ASR | 17.16 | 19.35 | 23.74 | 32.10 |

Table 5: Scalability checks beyond CUB/AwA. Values report CAT+ ASR reduction and clean accuracy when available.

| Dataset | Undefended Acc | ConceptGuard Acc | Undefended ASR | ConceptGuard ASR |
|---|---|---|---|---|
| LAD-E | 79.00 | 73.66 | 77.01 | 9.38 |
| LAD-V | 79.06 | 80.58 | 73.81 | 5.42 |
| CelebA | 77.02 | 76.88 | 82.14 | 18.65 |
| CheXpert | 82.75 | 82.50 | 79.35 | 14.22 |

## 6.6 Computation and Embedding Sensitivity

Tables 6 and 7 address two practical concerns. First, ConceptGuard does not multiply the full image backbone. Training of heads is parallelizable and inference overhead is a few milliseconds in the reported setting. Second, the defense is not tied to one unusually lucky embedding. BERT performs best here, but Word2Vec and CLIP-text also reduce ASR sharply relative to the undefended baseline.

## 7 Mechanism Analysis

The experiments support a specific mechanism: ConceptGuard works when partitioning keeps the number of effectively corrupted sub-models below the voting margin. This section connects the observed tables to that mechanism.

**Partial-trigger learnability.** An isolated sub-model does not necessarily memorize a target label from a small fraction of the trigger. On CUB/CAT+ with trigger size 20, forcing an isolated sub-model to observe 4, 8, 12, and 16 trigger concepts gives ASR values of 12.3%, 18.5%, 47.2%, and 75.8%, respectively. The sharp increase between 8 and 12 concepts is best interpreted as a CUB-specific partial-trigger memorization diagnostic, not as a universal threshold. The practical message is that a partition must keep the trigger density per vulnerable head below the dataset/model-specific memorization region.

**Corrupted sub-model count.** For CUB/CAT+ with $m = 4$ and trigger size 20, empirical $k$ measurements show that 94.2% of test samples have $k = 0$, 5.1% have $k = 1$, and only 0.7% have $k \geq 2$. The voting tolerance $\sigma$ is typically 1–2 in this setting. This explains why the ensemble remains stable even though some individual heads are imperfect: most samples do not produce enough corrupted votes to overcome the margin.

**Non-monotonic partition-size behavior.** Table 2 shows non-monotonic ASR as $m$ changes. This does not contradict the mechanism. Increasing $m$ changes both the number of concepts per head and the semantic boundary assignments. A larger $m$ can split trigger concepts more finely, but it can also create a cluster where several selected trigger concepts concentrate. This is why we report the full sweep rather than only the best value. The defensive recommendation is to treat $m$ as a validation hyperparameter and inspect scatter diagnostics, not to assume monotone improvement.

**Adaptive and correlated triggers.** Semantic clustering is useful against static attacks, but a partition-aware attacker can select concepts to affect multiple clusters. This raises adaptive ASR for semantic Con-

Table 6: Computational overhead on CUB. The expensive encoder is shared; ConceptGuard duplicates only lightweight heads.

| Method | Parameters | FLOPs | Training Time | Latency |
|---|---|---|---|---|
| Standard CBM | 23.6M | 4.12G | 2.50h | 18.2ms |
| ConceptGuard ($m = 4$) | 24.0M | 4.31G | 2.75h (+10%) | 21.5ms (+3.3) |
| ConceptGuard ($m = 6$) | 24.2M | 4.40G | 2.88h (+15%) | 23.1ms (+4.9) |
| ConceptGuard ($m = 8$) | 24.4M | 4.50G | 3.05h (+22%) | 24.8ms (+6.6) |

Table 7: Embedding and clustering ablation on CUB/CAT+ with $m = 4$.

| Clustering configuration | Clean ACC | ASR |
|---|---|---|
| Word2Vec + K-means | 77.82 | 22.45 |
| CLIP-text + spherical K-means | 77.10 | 17.50 |
| BERT-base + spherical K-means | **78.56** | **17.16** |

ceptGuard to 41.52%. Random-overlap partitioning reduces adaptive ASR to 19.82% by removing a fixed deterministic target for trigger selection. A correlated-trigger variant that selects all trigger concepts from the tightest semantic cluster yields ASR 21.34% under semantic partitioning and 19.55% under random overlap. These values show that randomization helps, while also reinforcing the theoretical boundary: random overlap is an empirical mitigation and needs separate theory.

**Visual encoder role.** The main implementation shares a visual encoder and isolates the lightweight concept-to-label heads. When the downstream poisoned label loss is allowed to backpropagate fully into the visual encoder, CUB/CAT+ ASR rises from 17.16% to 35.42%. This remains below the undefended 89.68% but shows that encoder isolation matters. In deployments where the encoder is trained end-to-end on potentially poisoned labels, the visual pathway itself becomes part of the attack surface.

**Cluster stability.** Across 10 clustering seeds, the cluster balance ratio is $1.31 \pm 0.08$. Balanced clusters are useful because they prevent a single head from receiving too many concepts and becoming both overly influential and more vulnerable. When balance or semantic stability is poor, random overlap or expert-defined partitions may be preferable.

## 8   Limitations and Scope

ConceptGuard is not a universal defense against all concept-space attacks. Its cleanest theoretical statement is conditional on the number of effectively corrupted sub-models and assumes disjoint partitions. If an attacker can force enough sub-models to flip, the majority vote can fail. Random-overlap partitioning improves empirical resilience against partition-aware attackers, but the theorem in Section 5 does not cover overlapping partitions.

The defense also depends on concept quality. Noisy learned concepts, biased concept predictors, or domain-specific vocabularies can reduce clean vote margins and make concept grouping less meaningful. Our noise stress tests show graceful degradation in the evaluated setting, not invariance to arbitrary concept noise. In high-stakes domains, concept partitions should be validated with domain-specific embeddings, expert-defined groups, or data-driven correlation analysis.

Finally, the partial-trigger memorization region is dataset- and architecture-specific. The CUB diagnostic suggests that small partial triggers are weak for the reported MLP heads, but other concept vocabularies, larger heads, higher poisoning rates, or stronger attackers may shift this region. The intended use of ConceptGuard is therefore diagnostic as well as defensive: practitioners should measure $k$, $\sigma$, trigger scatter, and partial-trigger behavior rather than relying on a fixed trigger-size rule.

## Broader Impact

This work is dual-use. The positive goal is to make interpretable models safer by addressing a failure mode that specifically exploits the concept interface. This is relevant in domains where CBMs are used because users care about explanations, interventions, or auditing. The risk is that a clearer analysis of concept-level backdoors and adaptive trigger selection could also help attackers design stronger attacks. We mitigate this by focusing on defensive mechanisms, diagnostic measurements, and limitations rather than operational attack recipes beyond the already established CAT/CAT+ threat model. The paper does not claim that ConceptGuard makes high-stakes deployments safe by itself; it identifies a concrete defense baseline and the conditions under which it should be trusted.

## 9 Conclusion

This paper studies defense against concept-level CBM backdoors under a threat model established by prior TMLR work. ConceptGuard partitions the concept space, trains subgroup concept-to-label heads, and aggregates predictions by majority vote. The mechanism reduces CAT/CAT+ attack success in evaluated settings while preserving clean utility, and its behavior can be understood through corrupted sub-model count, voting margin, partial-trigger learnability, and trigger scatter. The central claim is deliberately scoped: disjoint partitions support a conditional voting-margin analysis, while random overlap is an empirical adaptive mitigation. This makes ConceptGuard a transparent defense baseline for secure CBMs and a diagnostic framework for future work on adaptive semantic backdoors.

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

# A  Additional Theory and Proof Details

## A.1  Vote-Margin Proof Ledger

The main text gives the formal statement. Here we spell out the vote-count accounting and its relation to the original ECCV derivation. Let $c$ be a clean concept vector and $c'$ be the triggered vector. Let

$$\mathcal{H}(c, c') = \{j : f^j(\mathcal{G}^j(c')) \neq f^j(\mathcal{G}^j(c))\}$$

be the set of heads whose predictions change, and let $k = |\mathcal{H}(c, c')|$. For every label $\ell$, the attacked vote count $N_\ell(c')$ satisfies

$$N_\ell(c) - k \leq N_\ell(c') \leq N_\ell(c) + k. \tag{16}$$

The lower bound is attained in the worst case when every changed head that originally voted for $\ell$ moves away from $\ell$; the upper bound is attained when every changed head that did not originally vote for $\ell$ moves to $\ell$. Let $y$ be the clean ensemble prediction. The attacked ensemble keeps prediction $y$ whenever

$$N_y(c') \geq \max_{\ell \neq y}(N_\ell(c') + \mathbb{I}(y > \ell)). \tag{17}$$

Using the worst-case lower bound $N_y(c') \geq N_y(c) - k$ and the worst-case upper bound $N_\ell(c') \leq N_\ell(c) + k$, it suffices that

$$N_y(c) - k \geq \max_{\ell \neq y}(N_\ell(c) + k + \mathbb{I}(y > \ell)). \tag{18}$$

Rearranging yields Equation 10. This argument is deterministic once the clean votes and corrupted-submodel count are fixed. Importantly, the proof does not need to assume that a training algorithm is robust to label noise. Any realized head change, whether caused by partial-trigger learning, poisoned labels, or visual-concept coupling, is absorbed into the measured value of $k$.

## A.2  Dataset-Level Lower-Bound Procedure

For a fixed possible corrupted group set $\mathcal{J}$, evaluate Equation 14 on every test sample and count the clean-correct samples for which the original ensemble winner remains stable under worst-case reallocation of the votes in $\mathcal{J}$. Equation 15 then takes the minimum over all $\mathcal{J}$ of a chosen size $k$. This is a pessimistic diagnostic because it assumes the same worst-case group set can be corrupted across the whole dataset.

1. Compute clean subgroup predictions and clean vote counts for each test sample.

2. Check whether the clean ensemble prediction is correct; only clean-correct samples contribute to the diagnostic lower bound.

3. For each candidate corrupted group set $\mathcal{J}$, remove from the clean winner the votes contributed by groups in $\mathcal{J}$.

4. Add worst-case votes from groups in $\mathcal{J}$ to each competing label as in Equation 14.

5. Count whether the original clean ensemble label still wins after deterministic tie breaking.

6. Return the minimum clean-correct stable accuracy over all candidate sets $\mathcal{J}$.

For CUB/CAT+ with $m = 4$ and the measured $k$ distribution, this procedure gives a 74.82% lower-bound diagnostic, close to the empirical defended clean accuracy of 78.56%. The gap is expected: the diagnostic uses worst-case vote reallocation, while empirical attacks may not always allocate corrupted heads to the strongest possible competing label.

### A.3 Relation to the ECCV Theorem 2 Derivation

The ECCV version presented the dataset-level result as an improved joint accuracy bound. The Hold-on TMLR version keeps the same vote accounting but deliberately changes the interpretation. It is a lower-bound diagnostic for fixed-size disjoint group corruption, not a certified robustness theorem for arbitrary adaptive triggers. This wording is narrower and better aligned with the meta-review concern: random-overlap partitioning and fully white-box adaptive selection require separate analysis because one triggered concept may influence several overlapping heads.

## B Attack Formulation Details

CAT/CAT+ attacks operate by selecting trigger concepts and operations that create a semantic pattern associated with the target class. For a dataset $\mathcal{D}$ and a selected concept operation, CAT+ estimates a target-correlation statistic. If $n$ is the total number of training samples and $n_{\text{target}}$ is the number of target-class samples, the initial target probability is $p_0 = n_{\text{target}}/n$. For a modified concept condition $c_a$, the conditional target probability is written as $p^{(\text{target}|c_a)}$. CAT+ selects concepts and operations that increase a score of the form

$$\mathcal{Z}(c_a) = \left[ p^{(\text{target}|c_a)} - p_0 \right] / \left[ \frac{p_0(1 - p_0)}{p^{(\text{target}|c_a)}} \right].$$

The trigger set is built iteratively until the desired trigger size is reached. The defense does not use this score; it is included to clarify why CAT+ produces stronger and more semantically correlated triggers than simple random concept selection.

## C Dataset and Concept Preprocessing Details

CUB attributes are provided as structured strings, such as `has_bill_shape::dagger`; these are rewritten into short readable phrases such as "Bill shape is dagger" before text embedding. AwA attributes are single-token tags, so they are expanded into short descriptions. This enrichment is a one-time preprocessing step for text embedding. All compared methods use the same concept vocabulary to avoid confounding the defense with vocabulary differences.

For example, `has_bill_shape::dagger` is embedded as "Bill shape is dagger" and `has_eye_color::black` as "Eye color is black". AwA tags receive short contextual expansions, e.g., `meat` becomes "The animal consumes meat as part of its diet" and `forest` becomes "The animal inhabits forests." These rewrites affect only the text used to form concept embeddings; the underlying concept labels and model inputs are unchanged.

## D Experimental Settings

All main experiments use a shared ResNet-50 encoder and lightweight MLP concept-to-label heads. For CUB, the batch size is 64. For AwA, the batch size is 128. Both use Adam with learning rate $10^{-4}$, weight decay $5 \times 10^{-5}$, and exponential learning-rate decay with $\gamma = 0.95$. The concept loss weight is 0.5. Training images use random color jittering, random horizontal flips, and random cropping to 256 pixels. Inference uses center crop and resize to 256 pixels. Experiments were run on an NVIDIA A800 GPU.

# E  Injection Rate and Trigger Size

Table 8: CUB performance under varying injection rates and trigger sizes. Middle four columns correspond to the default 5% injection setting.

| Trigger | 2% injection | | | | 5% injection | | | | 10% injection | | | |
| | CAT | | CAT+ConceptGuard | | CAT | | CAT+ConceptGuard | | CAT | | CAT+ConceptGuard | |
| | ACC | ASR | ACC | ASR | ACC | ASR | ACC | ASR | ACC | ASR | ACC | ASR |
|---|---|---|---|---|---|---|---|---|---|---|---|---|
| 12 | 80.72 | 13.97 | 82.34 | 21.69 | 78.70 | 24.05 | 80.07 | 34.49 | 74.66 | 38.08 | 76.35 | 44.69 |
| 15 | 80.22 | 11.94 | 82.27 | 14.68 | 78.08 | 22.97 | 79.79 | 38.46 | 74.02 | 38.72 | 74.87 | 43.84 |
| 17 | 80.31 | 25.07 | 82.29 | 31.94 | 78.86 | 46.69 | 80.45 | 38.88 | 73.27 | 61.28 | 75.77 | 31.21 |
| 20 | 80.20 | 30.33 | 82.36 | 11.78 | 78.01 | 44.66 | 78.56 | 17.16 | 73.85 | 60.48 | 75.89 | 34.21 |
| 23 | 80.31 | 20.42 | 81.88 | 22.38 | 78.06 | 32.48 | 79.10 | 42.37 | 72.63 | 47.02 | 76.87 | 40.49 |

The table shows that ConceptGuard remains useful across trigger sizes and injection rates, but higher poisoning rates and larger triggers can raise ASR. This supports the main-text decision to describe partial-trigger behavior as a measured diagnostic rather than a universal rule.

# F  Target-Class Sensitivity

Table 9: CUB target-class sensitivity at 5% injection and trigger size 20.

| Target | CAT | | CAT+ConceptGuard | | CAT+ | | CAT++ConceptGuard | |
| | ACC | ASR | ACC | ASR | ACC | ASR | ACC | ASR |
|---|---|---|---|---|---|---|---|---|
| 8 | 75.06 | 74.24 | 79.72 | 21.93 | 75.56 | 52.63 | 79.96 | 11.19 |
| 16 | 75.16 | 40.91 | 80.36 | 8.74 | 75.72 | 68.81 | 80.13 | 30.52 |
| 24 | 74.37 | 35.48 | 80.03 | 17.24 | 74.91 | 54.48 | 81.00 | 13.38 |
| 32 | 74.70 | 37.68 | 80.62 | 25.31 | 75.58 | 17.87 | 79.53 | 9.52 |
| 40 | 74.46 | 42.35 | 79.53 | 7.93 | 74.96 | 23.77 | 80.32 | 12.49 |
| 48 | 75.09 | 49.77 | 80.74 | 4.49 | 75.73 | 95.11 | 80.91 | 22.14 |
| 56 | 75.22 | 70.99 | 81.07 | 15.93 | 75.23 | 57.36 | 80.24 | 15.25 |
| 64 | 74.85 | 43.63 | 80.41 | 10.27 | 74.58 | 84.27 | 80.46 | 19.29 |
| 72 | 74.99 | 47.99 | 80.67 | 10.90 | 75.34 | 59.06 | 80.62 | 10.11 |
| 80 | 75.03 | 62.51 | 80.89 | 10.83 | 75.61 | 75.83 | 80.45 | 11.85 |
| 88 | 74.75 | 51.13 | 79.94 | 21.93 | 74.66 | 72.71 | 80.10 | 25.02 |
| 96 | 74.82 | 17.73 | 80.67 | 8.99 | 75.20 | 62.16 | 80.91 | 14.78 |
| 104 | 74.84 | 53.02 | 80.65 | 12.80 | 74.92 | 40.42 | 80.00 | 16.03 |

# G  Additional Cluster Sweep at Higher Injection Rate

Table 10: ASR under varying cluster counts at 10% injection on CUB.

| $m$ | CG(CAT) | CG(CAT+) |
|---|---|---|
| 1 | 60.48 | 92.40 |
| 3 | 41.67 | 57.46 |
| 4 | 38.50 | 34.21 |
| 5 | 29.55 | **28.24** |
| 6 | 29.55 | 28.78 |
| 7 | **23.70** | 49.01 |
| 8 | 35.27 | 51.72 |
| 9 | 24.13 | 31.35 |
| 10 | 25.87 | 43.93 |

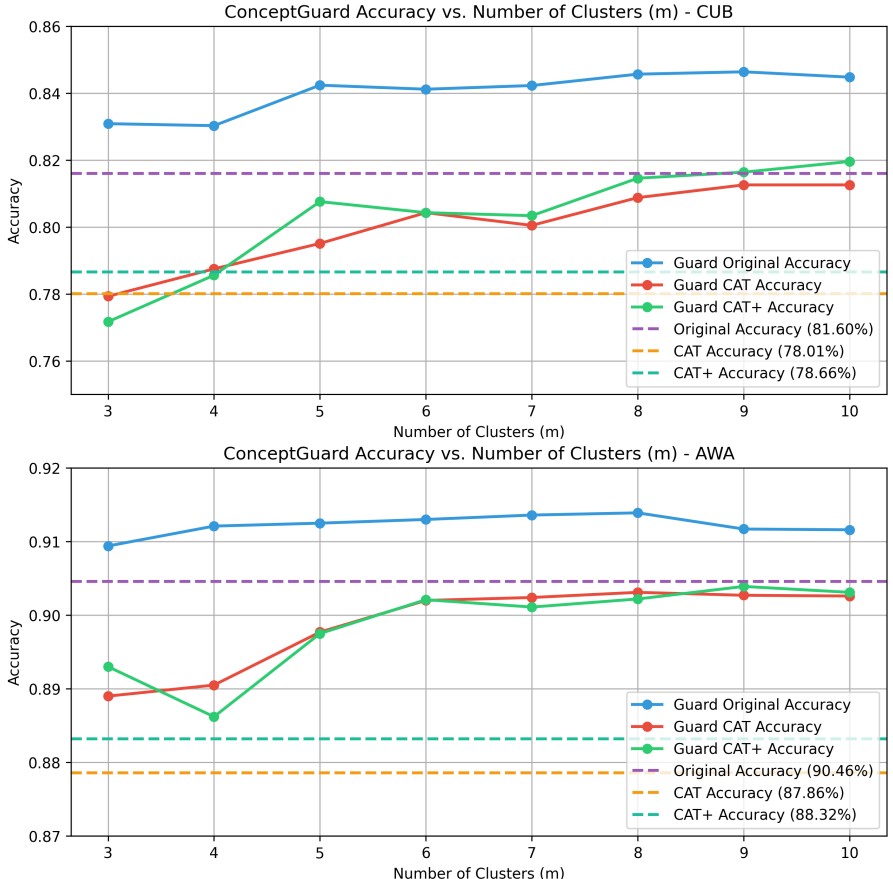

Figure 4: Clean-accuracy trend across cluster counts on CUB. ConceptGuard trained on clean data gives the upper-bound utility of the partitioned ensemble, while ConceptGuard under CAT+ shows the utility retained after poisoning. The gap narrows as $m$ increases, and the defended CAT+ model remains competitive with the undefended clean CBM at larger $m$.

## H    Individual Sub-Models and Ensemble Accuracy

Table 11: Accuracy of each CUB sub-model and the ensemble on clean test data.

|  | Original | CG(CAT) | CG(CAT+) |
|---|---|---|---|
| Base model 1 | 77.61 | 73.47 | 73.09 |
| Base model 2 | 78.49 | 73.97 | 74.02 |
| Base model 3 | 81.34 | 77.05 | 76.70 |
| Base model 4 | 77.67 | 72.30 | 72.01 |
| Average | 78.78 | 74.20 | 73.96 |
| Ensemble | 83.03 | 78.75 | 78.56 |

Table 12: Accuracy of each AwA sub-model and the ensemble on clean test data.

|  | Original | CG(CAT) | CG(CAT+) |
|---|---|---|---|
| Base model 1 | 88.67 | 87.34 | 87.50 |
| Base model 2 | 89.52 | 86.13 | 86.44 |
| Base model 3 | 89.79 | 86.82 | 86.49 |
| Base model 4 | 89.85 | 86.73 | 86.54 |
| Base model 5 | 88.88 | 86.79 | 87.02 |
| Base model 6 | 88.94 | 87.51 | 87.81 |
| Average | 89.28 | 86.89 | 86.97 |
| Ensemble | 91.30 | 90.20 | 90.21 |

The ensemble exceeds the average sub-model accuracy because subgroup errors are not identical. This matters for both clean utility and robustness: the same voting structure that filters clean errors also prevents a small number of corrupted heads from dominating the final prediction.

## I Broader Backdoor Threat Context

Input-level backdoors and concept-level backdoors differ in where the trigger lives. Input-level triggers are artifacts in the raw input, such as patches or rare tokens. Concept-level triggers are semantic patterns in the bottleneck representation. A concept-level trigger can be composed of plausible concepts, making it harder to detect by inspecting raw images alone. This is why input purification, trigger inversion, and model pruning are not sufficient substitutes for concept-space defenses.

## J Interpreting Baseline Defense Failures

The baseline table in the main text is not meant to imply that existing backdoor defenses are ineffective in their native threat models. The point is narrower: defenses designed for raw-input triggers or monolithic model repair are poorly aligned with a semantic trigger that lives in the CBM concept-to-label pathway.

**Input perturbation and filtering.** STRIP-style defenses perturb inputs and search for abnormal prediction consistency. This is well matched to some input-space triggers, but a concept-level backdoor can be activated by the model's predicted semantic pattern. If the input perturbation does not disrupt the concept pattern that the poisoned concept-to-label head uses, the defense has little leverage. In our setting, the visual trigger is only the mechanism that induces the semantic trigger; the decisive rule is learned downstream.

**Trigger inversion.** Neural Cleanse-style methods search for compact input-space triggers that cause a target class. In CBMs under concept-level attack, the trigger is a combinatorial pattern over discrete or thresholded concepts. Inverting such a trigger requires finding images that reliably induce a chosen semantic vector, which is a different optimization problem from finding a small patch. Even if a visual patch exists in the end-to-end poisoning setup, recovering that patch is not sufficient to characterize the semantic trigger learned by the concept-to-label classifier.

**Model pruning and repair.** Fine-Pruning assumes that backdoor behavior can often be associated with dormant or suspicious neurons. In a CBM, the trigger coordinates are legitimate concepts that also support clean predictions. Removing or suppressing them can reduce clean interpretability and utility. This is especially problematic for CAT+, where the trigger is selected to be plausible and correlated rather than obviously anomalous.

**Generic ensembles.** Bagging and random initialization improve diversity at the data or parameter level, but if every base classifier sees the full concept vector, every base classifier can still learn the full trigger. ConceptGuard changes the feature exposure itself. This is why the relevant comparison is not ensemble versus non-ensemble, but full-concept ensemble versus concept-partitioned ensemble.

**Robust aggregation.** Median aggregation and randomized smoothing-style baselines reduce the effect of some corrupted predictions, but they do not guarantee that semantic trigger evidence is isolated. If all heads observe the trigger, robust aggregation may aggregate consistently wrong predictions. ConceptGuard aims to make the wrong predictions sparse across heads by construction.

## K    Feature-Space Visualization Note

The original ECCV appendix included a t-SNE analysis of the visual feature space. The qualitative conclusion is that, under the shared-encoder configuration with blocked or concept-dominant label gradients, poisoned images do not form a clearly isolated malicious feature cluster. The final TMLR package does not rely on this visualization for a central claim; the central evidence is the quantitative ASR, $k$, and voting-margin diagnostics.

## L    ConceptGuard Procedure Details

The main text describes ConceptGuard at the level needed to understand the method. Here we spell out the implementation flow used in the experiments.

**Training procedure.** Given a potentially poisoned dataset $\mathcal{D}$, a concept vocabulary $\mathcal{C}$, and a desired number of groups $m$, the defender first embeds every concept name. For CUB, structured attribute strings are converted into short natural phrases before embedding. For AwA, single-token attributes are expanded into short descriptions so that sentence encoders have enough context. The defender then computes a partition map $\pi : \{1, \ldots, L\} \to \{1, \ldots, m\}$ for disjoint partitions, or a multi-map $\pi_{\mathrm{ov}} : \{1, \ldots, L\} \to 2^{\{1,\ldots,m\}}$ for overlapping partitions.

For each training example $(x_i, c_i, y_i)$, the full concept vector is split according to the partition map. The sub-vector assigned to group $j$ is paired with the same label $y_i$. Each head $f^j$ is trained on its subgroup dataset. The image encoder is shared, so the image is not processed independently by $m$ large backbones. This design is important for making the method practical: the ensemble is over concept-to-label heads, not over full image classifiers.

**Inference procedure.** At test time, the image is passed through the shared encoder once to obtain $\hat{c} = g(x)$. The predicted concept vector is split into subgroup vectors, each head predicts a label, and the final label is selected by majority vote. Ties are resolved by a deterministic lower-index rule, which is why the voting-margin theorem includes the tie-breaking indicator.

The implementation flow is therefore: embed concept names; form disjoint semantic groups or the random-overlap groups used only in the adaptive empirical variant; construct subgroup datasets by replacing each full concept vector with its assigned sub-vector; train the lightweight heads independently while sharing the expensive visual encoder; vote over subgroup predictions; and record the diagnostics used in the main paper, including clean voting margin, corrupted sub-model count $k$, trigger scatter, and partial-trigger ASR.

## M    Random-Overlap Partitioning Details

Random-overlap partitioning is used only as an empirical adaptive-defense variant. It is motivated by the following failure mode: if the attacker knows a deterministic semantic clustering assignment, then the attacker can intentionally choose trigger concepts that affect a majority of clusters. Randomizing the assignment makes the attacker's partition-aware selection less reliable when the exact random assignment is not known at trigger-selection time.

In the reported random-overlap setting, concepts are assigned to groups uniformly at random with controlled redundancy, so that each concept can appear in more than one subgroup. The overlap improves clean utility compared with purely random disjoint partitions because each head retains more semantic context. At the same time, randomness disrupts deterministic concentration strategies. This explains the empirical pattern

in Table 3: semantic disjoint partitioning is strongest under static CAT+, while random overlap is stronger under the adaptive selection protocol.

The important caveat is theoretical. Overlap means that sub-model errors are no longer independent or disjoint in the sense used by Theorem 1. A single triggered concept may affect multiple heads, and a single head may share concepts with another head. For this reason, the paper does not describe random overlap as covered by the disjoint-partition voting-margin theorem. A future proof would need to model overlap multiplicity, dependency among heads, and the probability that a trigger pattern maps to a majority of overlapping groups.

## N    Deployment Diagnostics

ConceptGuard should be deployed with diagnostics rather than a fixed trigger-size rule. The quantities that connect the theory and experiments are the clean voting margin $\sigma$, the corrupted sub-model count $k$, trigger scatter across groups, partial-trigger ASR, cluster balance, embedding sensitivity, and noise stress under learned or imperfect concepts. Together they test whether the empirical attack instance stays inside the regime described by the conditional voting-margin theorem.

These diagnostics also define failure modes. If the clean margin is small, even one corrupted head can change the decision. If trigger scatter is poor, the attack can fully corrupt one head and may corrupt several. If partial-trigger ASR is high, then splitting a trigger is not enough. If the partition changes drastically under embedding choices or random seeds, the defense should be paired with validation-based model selection or random overlap.

## O    Computational and Deployment Discussion

The main cost distinction is between total work and wall-clock time. Training $m$ heads creates more total work than training one head, but the heads are independent. If enough GPUs or CPU workers are available, the heads can be trained in parallel and the wall-clock time can be close to single-head training. The reported CUB timings reflect this shared-encoder design: $m = 4$ increases training time from 2.50 hours to 2.75 hours and latency from 18.2ms to 21.5ms per image.

Inference overhead is small because the encoder is evaluated once. The additional work is the forward pass through several shallow MLP heads and a vote count. This makes ConceptGuard more practical than an ensemble of complete CBMs, where each member would run a full image backbone. In deployments with strict latency budgets, the number of heads can be selected by validating clean accuracy, ASR under held-out attacks, and latency together.

Partition choices expose a practical trade-off. Smaller $m$ preserves more concept context inside each head, but a trigger can occupy more of that head's input. Larger $m$ lowers trigger density per head, but can reduce utility and create partition-boundary artifacts. Semantic disjoint clustering gives strong static performance, while random or overlap-based variants reduce predictable cluster structure at the cost of weaker clean-context control. Expert-defined groups are useful in specialized domains, especially when generic text embeddings misrepresent domain semantics.

## P    Additional Limitations

The main limitations section focuses on the theory-practice boundary. We add two operational limitations here.

**Concept vocabulary dependence.**    CBMs assume that the concept vocabulary is meaningful and sufficiently complete for the task. If the vocabulary omits important causal features, a concept-level defense can preserve an incomplete or biased decision process. ConceptGuard does not fix concept set design. It only changes how the existing concept representation is used by the label predictor.

**Potential collusion across sub-models.** An adaptive attacker could try to construct triggers whose partial patterns are individually weak but jointly steer several heads toward the same target. Majority voting is vulnerable if enough heads align on the same wrong label. Detecting such collusion requires measuring not only whether heads flip, but also whether their flipped labels concentrate on the attack target. This is why the $k$ diagnostic should be label-aware in adaptive evaluations.

**Domain-specific embeddings.** General-purpose embeddings may be poorly calibrated for specialized vocabularies. In medical settings, two concept names may be linguistically similar but clinically distinct, or vice versa. Domain-specific encoders such as biomedical text encoders, expert partitions, or concept co-occurrence graphs are natural replacements for generic BERT clustering. The voting-margin analysis remains applicable to any fixed partition; what changes is the empirical value of the margin and the measured trigger scatter.

