# OpenReview forum: "Guarding Concept Bottlenecks: A Voting-Margin Defense Against Concept-Level Backdoors"
_TMLR — Under review for TMLR_

### Review · Reviewer_eF3j · 2026-07-04

**Summary Of Contributions:**

This paper studies the problem of concept-level backdoor attacks in Concept Bottleneck Models. To mitigate such attacks, the authors propose a concept grouping and majority-voting strategy, and provide theoretical analysis to support the proposed approach.

Strengths:
1. The proposed method is novel and interesting. The idea of using concept grouping together with majority voting provides a potentially useful direction for improving the robustness of Concept Bottleneck Models against concept-level backdoor attacks.
2. The theoretical analysis gives an intuitive explanation of why grouping and voting can be beneficial, and helps clarify the motivation behind the proposed defense.

Weaknesses:
1. My main concern is whether the proposed grouping strategy may hurt the clean-task performance. Although concept grouping may help defend against attacks, it may also make each individual classifier harder to train or less accurate, since each classifier only receives a subset of concepts. Therefore, the way concepts are grouped appears to be crucial. The paper would be stronger if it provided a more detailed analysis or additional experiments on the trade-off between robustness and clean accuracy, as well as the sensitivity to different grouping strategies.

2. The theoretical results are somewhat intuitive and relatively straightforward. While they help explain the high-level benefit of majority voting, the analysis seems to lack deeper insights into when the proposed method succeeds or fails. A more detailed theoretical characterization of the assumptions, limitations, and failure cases would make the contribution more convincing.

3. The models used in the experiments appear to be relatively dated. It would be helpful to discuss whether concept-level backdoor attacks and the proposed defense are applicable to more recent models or more challenging tasks. Additional experiments on stronger or more modern backbones/tasks would better demonstrate the practical relevance and generality of the method.

**Audience:**

Yes

**Audience Explanation:**

The paper proposes a method for defending against attacks, which is of practical and research interest to at least some members of the TMLR audience.

**Claims And Evidence:**

Yes

**Claims Explanation:**

Yes. The evidence presented in the paper generally supports the claimed advantages of the proposed method. However, I believe that some additional experiments are still needed to support several important aspects that are not fully addressed in the current submission.

**Requested Changes:**

Critical changes required for acceptance:

I believe the paper should analyze whether the proposed method leads to any performance degradation on standard/general tasks. In addition, the authors should provide a more in-depth analysis of the way concepts are grouped, as well as the impact of different grouping strategies.

Recommended changes to strengthen the work:

The paper could be strengthened by evaluating the method on more recent models and tasks. That said, if experiments in this area are commonly conducted on relatively older models, I think the current choice is still understandable.

---

### Review · Reviewer_Pukh · 2026-07-17

**Summary Of Contributions:**

This paper studies defenses against concept-level backdoor attacks on Concept Bottleneck Models (CBMs), following the recently proposed CAT/CAT+ threat model. The authors propose ConceptGuard, which partitions the concept space into multiple semantic groups, trains lightweight concept-to-label classifiers independently on each partition, and aggregates predictions through majority voting. The paper further provides a conditional voting-margin analysis explaining when the ensemble prediction remains stable under a bounded number of corrupted sub-models, together with several diagnostic quantities including corrupted-head count and trigger scatter. Extensive experiments are conducted on multiple CBM benchmarks under both static and adaptive attacks, showing substantial reductions in attack success rate while largely preserving clean accuracy.

Strength
1. Addresses an emerging security problem for CBMs that has received limited attention.
2. Simple and interpretable defense mechanism.
3. Experimental evaluation is relatively comprehensive, including adaptive attacks, noisy concepts, scalability studies, and computational overhead.
4. The paper explicitly scopes its theoretical claims and discusses limitations.

Weakness
1. The theoretical contribution is relatively limited and largely conditional on the corrupted-head assumption.
2. The novelty over existing partition-based or ensemble defenses could be better clarified.
3. Several empirical assumptions (e.g., semantic clustering, partial-trigger learnability) are mainly justified empirically rather than theoretically.

**Audience:**

Yes

**Audience Explanation:**

I believe this work would be of interest to the TMLR audience. Security of Concept Bottleneck Models is a relatively new research direction, and this paper focuses on defending against a recently proposed threat model. Even if the proposed defense is relatively simple, the empirical findings and the diagnostic perspective on concept-space partitioning provide useful insights for future research on secure concept-based models.

**Broader Impact Concerns:**

I do not have major broader impact concerns beyond the standard dual-use nature of security research. The paper already discusses the potential misuse of stronger attack understanding and appropriately frames the work as a defensive contribution.

**Claims And Evidence:**

Yes

**Claims Explanation:**

Overall, I believe the main empirical claims are supported by the presented experiments. The proposed defense consistently reduces attack success rates across multiple datasets, attack settings, and several ablations while largely maintaining clean accuracy. The paper also evaluates adaptive attacks, noisy concepts, embedding choices, and computational overhead, providing a fairly comprehensive empirical study.

However, I believe several theoretical claims should be interpreted carefully. The proposed theorem establishes a conditional voting-margin guarantee assuming the number of effectively corrupted sub-models is bounded, rather than explaining why the proposed partitioning strategy necessarily leads to such bounded corruption. While the authors explicitly acknowledge this limitation, the theoretical analysis mainly serves as an interpretation of the observed mechanism instead of providing a robustness guarantee for the proposed training procedure.

**Requested Changes:**

The first issue I would like the authors to strengthen is the positioning of the proposed method with respect to existing ensemble and partition-based learning approaches. While the paper clearly demonstrates that generic full-concept ensembles are insufficient for defending against concept-level backdoors, it is less clear what aspects of the proposed concept-space partitioning are fundamentally new beyond existing feature-space partitioning or ensemble learning strategies. A more detailed discussion of the relationship to prior partition-based ensembles, random subspace methods, or other feature-level robustness techniques would help clarify the novelty and better position the contribution within the broader literature.

Second, I encourage the authors to further clarify the role and scope of the theoretical analysis. The current voting-margin theorem provides a conditional robustness guarantee once the number of effectively corrupted sub-models is bounded, but it does not explain why the proposed semantic partitioning is expected to produce such a favorable corruption pattern. Although the paper explicitly acknowledges this limitation, I believe the connection between semantic partitioning and the resulting corrupted-head distribution could be better justified, either through additional analysis or a more careful discussion of the underlying assumptions.

Finally, while the empirical evaluation is comprehensive, the adaptive attack evaluation could be strengthened. The current adaptive setting primarily considers partition-aware trigger selection, whereas stronger white-box adaptive strategies that jointly optimize against the ensemble remain largely unexplored. Evaluating stronger adaptive attackers, or providing a more detailed discussion of their expected behavior and limitations, would further strengthen the empirical evidence supporting the proposed defense.

Minor revisions that would further improve the paper include providing additional justification for the choice of semantic text embeddings and clustering methods, clarifying the practical considerations when selecting the number of partitions, and improving several presentation details to further distinguish theoretical guarantees from empirically observed mechanisms.

---

### Review · Reviewer_Eq1n · 2026-07-21

**Summary Of Contributions:**

This paper proposes ConceptGuard, a defense against concept-level backdoor attacks on Concept Bottleneck Models. The method partitions concepts into several groups, trains separate concept-to-label classifiers, and aggregates their predictions through majority voting. The paper also provides a conditional voting-margin analysis and evaluates the method under static and adaptive CAT/CAT+ attacks. The results show substantial reductions in attack success rate while generally preserving clean accuracy. The idea is intuitive and the experiments are relatively comprehensive, although the theoretical guarantee is conditional and does not fully cover the random-overlap variant.

**Audience:**

Yes

**Audience Explanation:**

Concept-level backdoors and the security of Concept Bottleneck Models are relevant to researchers in interpretable and trustworthy machine learning. However, the current technical contribution may be of limited interest because the method mainly combines concept partitioning with standard ensemble voting, and the theoretical analysis does not substantially explain when the required robustness condition will hold.

**Broader Impact Concerns:**

The broader impact discussion is generally adequate.

**Claims And Evidence:**

No

**Claims Explanation:**

The experiments demonstrate effectiveness under the evaluated settings, but the central robustness argument relies on the assumption that only a small number of sub-models are corrupted. The theoretical result characterizes stability once this assumption already holds, rather than proving that ConceptGuard enforces it. In addition, performance degrades considerably under a partition-aware adaptive attack, while the random-overlap variant used to mitigate this issue is supported only empirically and is not covered by the analysis.

**Requested Changes:**

The paper should provide a stronger justification for why concept partitioning is expected to limit the number of corrupted sub-models, rather than treating this as an empirically measured condition. A more informative theoretical analysis connecting trigger properties, partition construction, and robustness would strengthen the contribution. The authors should also evaluate stronger fully adaptive attackers that know the complete defense configuration and better explain the novelty over standard feature-partitioning and ensemble defenses.